# Multiple imputation of maritime search and rescue data at multiple missing patterns

**Guobo Wang** **, Minglu Ma, Lili Jiang\*, Fengyun Chen, Liansheng Xu**

China Waterborne Transport Research Institute, Beijing, PR China

\* jianglili@wti.ac.cn

**Data Availability Statement:** All relevant data are within the manuscript.

**Funding:** This work was supported by the [National Science and Technology Support Program] under Grant [2015BAG20B01]; [National Key R&D

## Abstract

Based on the missing situation and actual needs of maritime search and rescue data, multiple imputation methods were used to construct complete data sets under different missing patterns. Probability density curves and overimputation diagnostics were used to explore the effects of multiple imputation. The results showed that the Data Augmentation (DA) algorithm had the characteristics of high operation efficiency and good imputation effect, but the algorithm was not suitable for data imputation when there was a high data missing rate. The EMB algorithm effectively restored the distribution of datasets with different data missing rates, and was less affected by the missing position; the EMB algorithm could obtain a good imputation effect even when there was a high data missing rate. Overimputation diagnostics could not only reflect the data imputation effect, but also show the correlation between different datasets, which was of great importance for deep data mining and imputation effect improvement. The Expectation-Maximization with Bootstrap (EMB) algorithm had a poor estimation effect on extreme data and failed to reflect the dataset's variability characteristics.

## 1 Introduction

Driven by the development of society and economy, the number and tonnage of ships have increased rapidly with the growth of shipping [1]. At the same time, problems such as shipping lane congestion, low standardization and ship aging have become increasingly serious. Additionally, the navigation environment has become increasingly complex and there has been a corresponding rise in water traffic safety risks [2,3]. In 2018, China had a total of 176 water traffic accidents with general and higher levels associated with transportation ships, causing 237 people missing, 83 shipwrecks, and a direct economic loss of 290 million yuan [4,5]. Water traffic accidents have significantly impacted the sustainability and stability of the water transportation economy. To ensure water traffic safety, minimize human casualties and property damage, it is essential to carry out maritime search and rescue (SAR) efficiently and effectively. The focus on maritime SAR research, conducted in main countries at present, is mainly the construction of SAR capability evaluation models and SAR systems [6–8]. To analyse Zhoushan's maritime SAR capabilities quantitatively, Liu et al. (2018) established a fuzzy comprehensive evaluation theory, and summarized the factors affecting SAR capabilities. Liu et al. also constructed an evaluation model, and the achieved results provided suggestions for

Program of China] under Grant
[2017YFC1404705].

**Competing interests:** The authors have declared
that no competing interests exist.

Zhoushan SAR [9]. Regarding the randomness and ambiguity of the evaluation index for the
maritime SAR emergency capability system, Zhang et al. (2019) proposed a new methodology
based on based on subjective-objective attribute weights integrated and centre normal grey
cloud method, and conducted model evaluation on three ports. The results showed that the
IAHP-CRITIC weights combination method could effectively integrate subjective and objec-
tive weight information [10]. On the basis of a detailed analysis of the maritime SAR process at
the sea area of Zhoushan, Wang et al. (2016) summarized the factors affecting the maritime
SAR capacity and conducted a fuzzy comprehensive evaluation of maritime SAR capacity of
the sea area in Zhoushan based on fuzzy mathematics, providing a quantitative basis for the
improvement of the SAR program [11]. Xu et al. (2014) proposed a maritime SAR capability
evaluation algorithm based on the cloud model. Experimental results showed that the method
was objective and accurate [12]. Weihong (2009) used the marine risk classification ontology
as a sample and proposed a reference model for marine SAR decision-making ontology, which
solved the acquisition, sharing, and reuse of heterogeneous data [13]. At present, existing stud-
ies have mainly analysed data on water traffic safety conditions and SAR capabilities. Subject
to regional disparity and economic conditions, there are differences in data completeness in
various regions. As a result, studies on maritime SAR capabilities are limited on specific
regions. Existing data cannot meet the simulation needs of the SAR model. The determination
of various parameters in the SAR model also tends to use subjective data due to data missing,
which brings substantial uncertainty to evaluation models.

Scientists around the world have attempted to solve the problem of model simulation diffi-
culty caused by missing data. Nieh et al. (2014) evaluated the imputation effect of multiple
imputation methods, deletion methods and mean value imputation methods on missing micro-
bial water quality data. The linear regression model of bacillus density was used to predict the
density of somatic coliphages. The results indicated that the multiple imputation method intro-
duced the minimum deviation compared with other methods [14]. Audigier et al. (2018) pro-
posed and compared multiple imputation methods for multivariate continuous and binary
data. The comparison showed that the relative performance of multiple imputation methods
varied with data missing patterns, multi-layer structures, and the types of missing variables [15].
Rawlings et al. (2017) used multiple estimation based on multiple imputation by chained equa-
tions (MICE) to estimate the cognitive performance scores of individuals who did not partici-
pate in the 2011–2013 atherosclerosis risk test. Studies have shown that MICE can be an
effective tool to estimate cognitive performance and improve cognitive decline assessment
when data are available for unreviewed individuals [16]. Fawzi et al. (2016) proposed a new
automatic and adaptive algorithm for selecting transformations of data-enhanced samples.
Results from this work showed that the algorithm was superior, in terms of accuracy and
robustness, to random data enhancement, and the results were comparable or higher than the
existing selective sampling methods [17]. Munoz-Bulnes et al. (2017) adopted loop detector vol-
ume data collected on Interstate 5 in Washington State. The researchers applied the iterative
multiple imputation method based on a chain equation, considered the spatial correlation
between nearby detectors in prediction and efficiently handled the presence of missing data in
all predictive variables. The results showed that the proposed method could outperform basic
pairwise regression and produce reliable interpolation estimates even if the data was missing for
the whole days or several months [18]. Vitale et al. (2019) combined the classic Expectation-
Maximization (EM) algorithm and Bootstrap method to interpolate the half-hour time series
data of the net carbon dioxide ecosystem exchange (NEE), latent heat flux (LE) and sensible
heat flux (H). The results from this research revealed that the model was better capable in terms
of complex dynamics of ecosystem flux and reconstruction of missing data points, providing
unbiased inferences and preserving the original sampling distribution [19].

Although many applications of data multiple imputation methods are conducted in China and abroad, there was no application in maritime SAR. Therefore, according to the scarcity conditions of each data type, this paper conducted multiple imputation simulations of data under different scarce conditions, endeavoring to give a better solution of the data shortcomings, provide a better understanding of the rules and features of SAR, improve the scientificity, foreseeability, initiative, and creativity of rescue work, and also solve prominent problems and key issues affecting the stability of Chinese water security and shipping safety.

## 2 Materials and methods

### 2.1 Data sources

The data utilized in this study were mainly used to analyze the common situation of maritime SAR, including national waterborne transport volume, bad weather, water traffic danger, ships and personnel in distress, and shipwrecks. The data sources are shown in Table 1. Since the weather affecting the safety of search and rescue at sea is mainly typhoon, the number of bad weather in the study is number of typhoon days.

### 2.2 Multi-imputation methods

**2.2.1 Multi-imputation principle.** The multi-imputation is a statistical analysis method for analysing missing data. The core idea of multi-imputation is to generate multiple imputed values by imputation method for missing data in the sample datasets, and then to construct multiple complete sample datasets. The complete dataset generated by the imputation is statistically analysed, and finally the expected value and variance of the sample are obtained; this process is summarized in three steps, including imputation, analysis, and merging. The analysis and merging process are as follows:

$$\bar{Q} = \frac{1}{m}\sum_{i=1}^{m}\hat{Q}^{(i)} \tag{1}$$

Where m is the number of data sets generated by imputation, $\hat{Q}^{(i)}$ is the ith imputation dataset, $\bar{Q}$ is the average of the imputed dataset.

$$B = \frac{1}{m-1}\sum_{i=1}^{m}(\hat{Q}^{(i)} - \bar{Q})^2 \tag{2}$$

Where B is the variance within the imputation set, i.e. the variance within the group.

$$\bar{U} = \frac{1}{m}\sum_{i=1}^{m}\hat{U}^{(i)} \tag{3}$$

**Table 1. Data sources for the analysis.**

| Data type | Data (2014.5–2019.8) (unit) | Source |
|---|---|---|
| Economic indicators | Waterborne transport volume ($10^4$ tons) | http://www.mot.gov.cn/ |
| Weather | Days of bad weather (day) | http://typhoon.nmc.cn/web.html |
| Security status | Times of distress (time) | http://zizhan.mot.gov.cn/sj2019/soujiuzx/ |
| | Number of ships in distress (ship) | |
| | Number of shipwrecks (ship) | |
| | Number of people in distress (person) | |
| | Number of dead or missing persons (person) | |

Where $\hat{U}^{(i)}$ is the square of the standard error of the ith imputation set, $\bar{U}$ is the variance between the imputation sets, i.e. the variance between the groups.

$$T = (1 + \frac{1}{m})B + \bar{U} \tag{4}$$

Where $T$ is the total variance estimate of the imputation set.

**2.2.2 DA algorithm.** DA algorithm (Data Augmentation) is a special type of MCMC method and its algorithm is divided into two steps: In the first step, with the sample data and the parameters given in the previous step for iteration, a sample is randomly selected from the posterior distribution of the missing data for the next calculation:

$$Y_{miss}^{(t+1)} \sim f(Y_{miss}|Y_{obs,}\theta^{(t)}) \tag{5}$$

In the second step, with the observation sample and the missing sample data extracted in the previous step given, a sample is randomly extracted from the posterior distribution of the parameter for the next iteration:

$$\theta^{(t+1)} \sim f(\theta|Y_{obs,}Y_{miss}^{(t+1)}) \tag{6}$$

In the first iteration of the beginning, $(Y_{miss}^{(0)}, \theta^{(0)})$, which is the initial value of $(Y_{miss},\theta)$ needs to be determined first.

After the first step of the calculation and the iteration of the second step, a Markov chain $\{(Y_{miss}^{(1)}, \theta^{(1)}), (Y_{miss}^{(2)}, \theta^{(2)}), \cdots\}$, that converges to the joint posterior distribution $f(\theta,Y_{miss}|Y_{obs})$ of $(Y_{miss},\theta)$ is obtained. Through the kth iteration, if the Markov chain satisfies the convergence condition, a $\theta^*$ can be randomly extracted from the posterior distribution of the parameter $\theta$. The missing values are extracted according to the obtained parameters to complete a single imputation, and the same steps are repeated m times to obtain m imputed complete datasets.

**2.2.3 EMB algorithm.** The EMB algorithm is a combination of the EM algorithm (Expectation-Maximization) and the Bootstrap algorithm. There are two basic assumptions to be met in applying this algorithm: (1) Interpolated datasets subject to multivariate normal distribution, i.e. $Y \sim N_p(\mu,\Sigma)$, Where Y is the $n \times p$-dimensional sample dataset, consisting of two parts, the missing data and the observation data, i.e. $Y = (Y_{obs},Y_{miss})$; (2) The response mechanism for missing data is a random missing, i.e. $f(M|Y,\phi) = f(M|Y_{obs},\phi)$, where $M = (m_{ij})_{n \times p}$ is the data missing mode matrix. When the elements in the sample dataset $y_{ij} \in Y_{miss}$, then $m_{ij} = 1$; otherwise, when $y_{ij} \in Y_{obs}$, $m_{ij} = 0$. $\phi$ is a parameter that determines the distribution of the data missing mode M. Under the conditions of the two basic assumptions above, let $\theta = (\mu,\Sigma)$, the likelihood function $L(\theta,\phi|Y_{obs},M) = f(Y_{obs},M|\theta,\phi)$ of $\theta$ and $\phi$, be decomposed as follows:

$$f(Y_{obs}, M|\theta, \phi) = f(Y_{obs}|\theta)f(M|Y_{obs}, \phi) \tag{7}$$

At this point, the maximum likelihood estimate of the parameter $\theta$ is completely dependent on $f(Y_{obs}|\theta,\phi)(L(\theta|Y_{obs}) \propto f(Y_{obs}|\theta))$.

When the prior distribution of the parameter $\theta$ is a prior distribution without information, then

$$f(\theta|Y_{obs}) \propto f(Y_{obs}|\theta) = \int f(Y|\theta)dY_{miss} \tag{8}$$

The acquisition of the posterior distribution $f(\theta|Y_{obs})$ of parameter $\theta$ represents the main difficulty in analysing the incomplete sample dataset. The posterior mode estimates of the

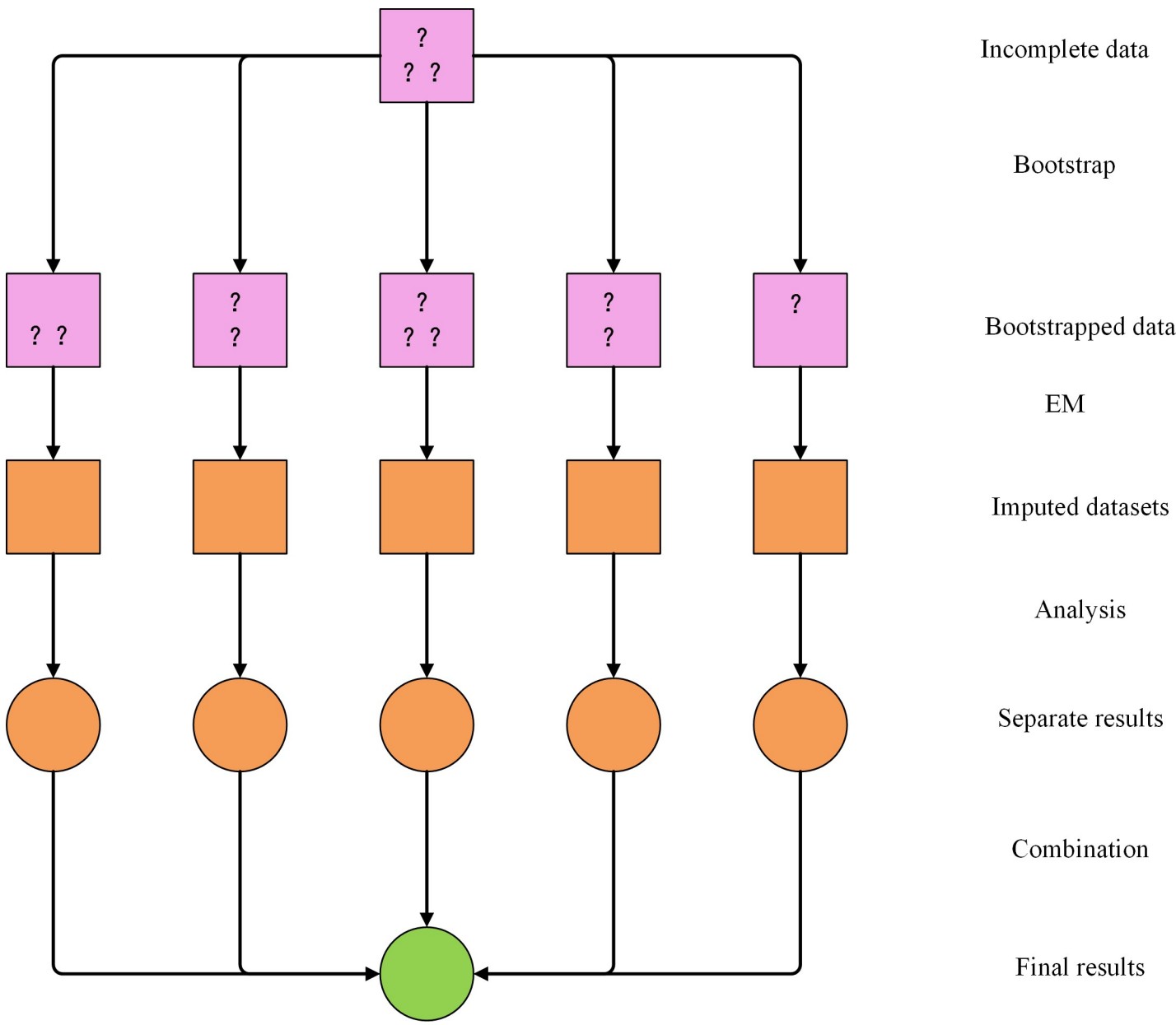

**Fig 1. EMB algorithm calculation process.**

parameters can be obtained using the EM algorithm, while the EMB algorithm is based on the EM algorithm and uses the Bootstrap method to sample the posterior distribution $f(\theta|Y_{obs})$.

Fig 1 shows the main principle process of the EMB algorithm. The first step is to combine the bootstrap algorithm to sample the missing data samples to obtain m missing data sample sets. Then, the EM algorithm is used to perform posterior mode estimation on the parameters of the m missing data sample sets. The third step is to calculate the expected value of the corresponding missing data and to interpolate to the missing position. Finally, statistical analysis of all samples provides the expected total variance of each imputed dataset. In Fig 1, each color represents a kind of dataset. Exactly, purple represents the incomplete dataset, and orange represents the complete dataset. Each graphic represents an interpolation process, the square

represents the process before the completion of the interpolation analysis, and the circle represents the process after the completion of the interpolation analysis.

## 2.3 Missing mode construction

This paper examines the national water safety situation and maritime SAR data as the analysis object, and discusses the lack of national maritime SAR data in time. The specific data missing situation is displayed in Fig 2. Wherein, Death is the number of missing or dead persons, PR is the number of people in distress, Shipwreck is the number of shipwrecks, SR is the number of ships in distress, TR is the times of distress, Weather is the number of bad weather, Volume is the volume of waterborne transport, and DATE is the calendar date. Fig 2 illustrates that the lack of data related to the national water safety situation and maritime SAR are mainly based

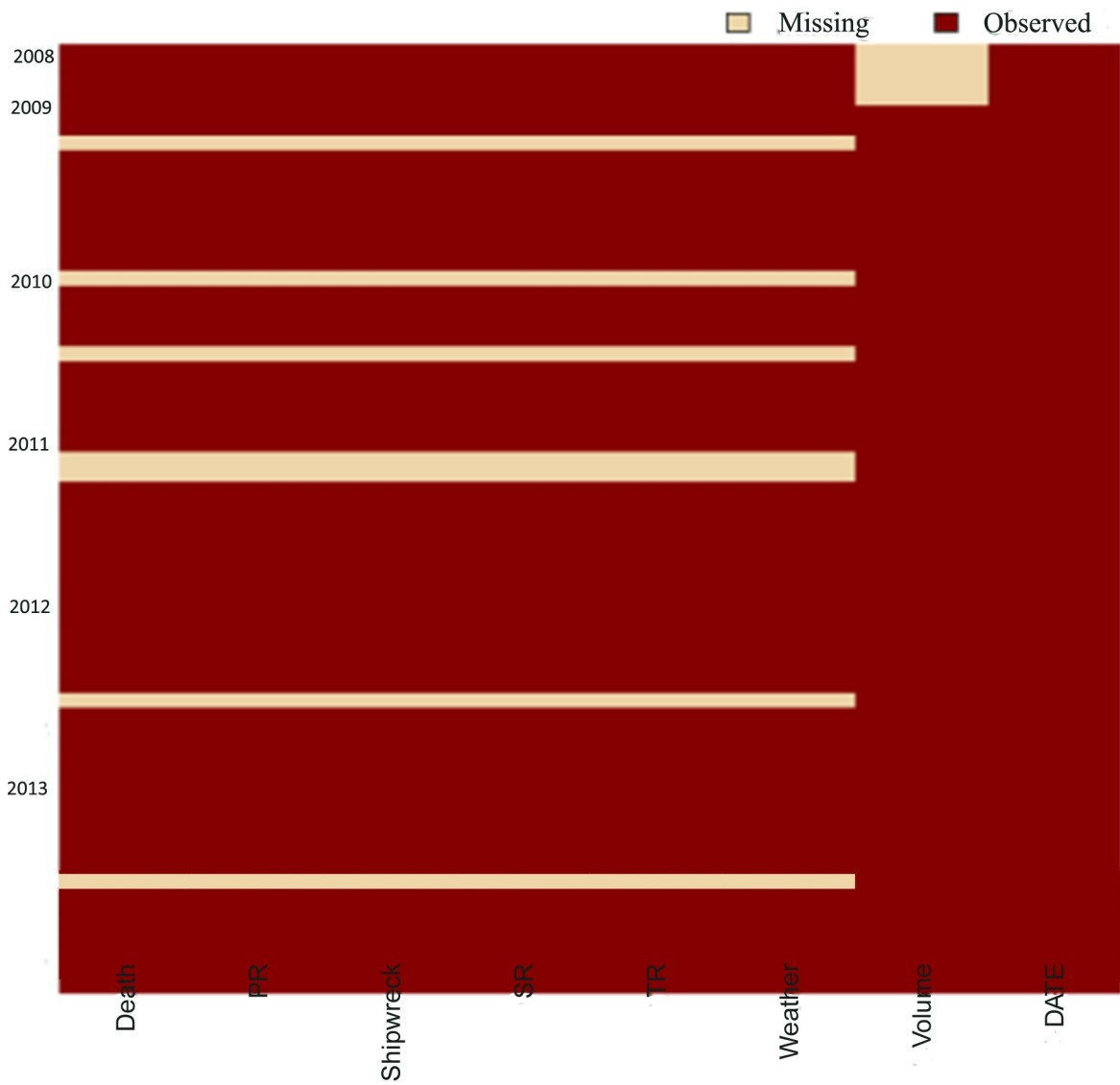

**Fig 2. Missing situation of different data types.**

on random continuous missing pattern. For example, data on water economy development, such as waterborne transport volume, has been continuously missing from January to May 2014. Data indicating the safety situation and SAR situation, such as the times of distress, the ship in distress, the rescued ship and the rescued person, have missing data with shorter time series in different years. In general, the lack of data related to the national water safety situation and maritime SAR are mainly based on random continuous missing, and with some short time series missing. Therefore, this paper takes the national waterborne transportation volume, water safety situation and maritime SAR data from May 2014 to August 2019 as the analysis object, and sets the data missing mode with a different missing rate.

The data missing patterns under different missing rates were mainly continuous with a small number of short time series missing data, which was consistent with the actual situation of the missing data, highlighting the practical significance of the research. To specifically analyse the impact of missing positions in different time series on the imputation results, different scenarios of missing positions were established to make the analysis of the missing data more comprehensive. Further, to better analyse the influence of different missing rates on the imputation results and reduce the impact of different missing position on the results, the included missing patterns were set; i.e., the missing set with lower missing rates was a subset of the missing set with higher missing rates. Fig 3 shows the specific missing data distribution pattern.

## 3 Imputation result analysis

The data of four different missing rates were interpolated using the DA algorithm and the EMB algorithm, and the imputation results were evaluated by standard deviation. Tables 2 and 3 show the data imputation effects under the four different missing rates. As the amount of missing data increases, the difference between the standard deviation of the imputation set and the standard deviation of the unmissed dataset gradually increases vis-à-vis the DA algorithm and the EMB algorithm. There was a difference in the degree of deviation after imputation of different data; that is, the imputation effects of different data types were different.

In Table 2, the DA imputation algorithm was used for data imputation, and the overall imputation effect was better, especially when the missing rate did not exceed 20%, the standard deviation after imputation of different types of datasets was very close to the standard deviation of the original dataset, and the imputation effect was ideal. When the missing rate exceeded 20%, the imputation effect significantly degraded, and the degree of impregnation of different types of data was different. Compared with other data types, the decline in the interpolating effect of the Volume, SR and Death was more pronounced. Compared with the DA algorithm, the dataset obtained by the EMB algorithm was clearly more advantageous, and the imputation results were closer to the original value. Moreover, even at higher data loss rates, the EMB algorithm maintained good imputation effect. From the imputation effect of different data types, the interpolating effect of Volume, TR and Death in different missing rates was highly significant, which was essentially consistent with the standard deviation of the original value. In comparison, the imputation effect of SR and PR was significantly reduced after the missing rate exceeded 20%. This showed that the two groups of data were less discrete. When the data were missing, the degree of dispersion changed considerably, which led to a certain deviation between the imputed data and the actual data.

There were also large differences in the complete datasets obtained using the DA algorithm and the EMB algorithm at different missing positions. Table 4 shows that the effect of the datasets imputed by the DA algorithm at different missing positions was generally better. However, the impact on the effect of imputation at different missing positions was distinct. Considering

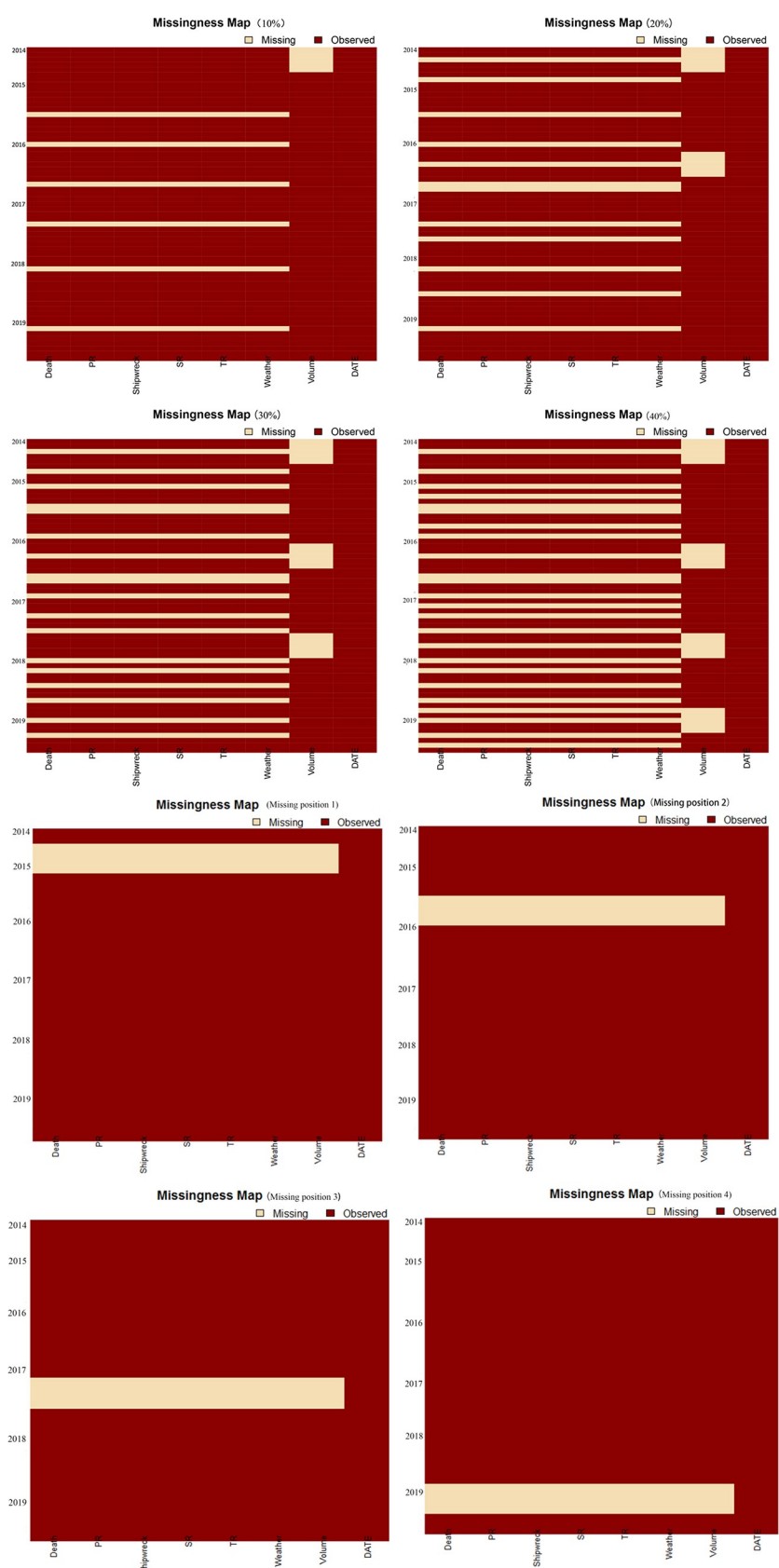

**Fig 3. Setting of different data missing patterns.**

**Table 2. Imputation effect of DA algorithm under different missing rates.**

| Missing rate | Volume | Weather | TR | SR | Shipwreck | PR | Death |
|---|---|---|---|---|---|---|---|
| 0% | 6438.993 | 5.44135 | 37.1087 | 31.0058 | 8.225633 | 353.1576 | 56.87932 |
| 10% | 6470.66 | 4.99744 | 51.4778 | 30.1941 | 8.89835 | 329.53 | 60.5553 |
| 20% | 5983.11 | 5.49071 | 51.209 | 35.0239 | 9.48112 | 293.808 | 64.2646 |
| 30% | 7427.82 | 6.64588 | 51.7267 | 25.0894 | 8.07559 | 314.527 | 65.7269 |
| 40% | 7618.53 | 5.99057 | 44.3055 | 26.3287 | 8.51698 | 412.911 | 87.4418 |

**Table 3. Imputation effect of EMB algorithm under different missing rates.**

| Missing rate | Volume | Weather | TR | SR | Shipwreck | PR | Death |
|---|---|---|---|---|---|---|---|
| 0% | 6438.993 | 5.44135 | 37.10876 | 31.00578 | 8.225633 | 353.1576 | 56.87932 |
| 10% | 6442.706 | 5.35331 | 35.33067 | 29.08767 | 7.967874 | 334.6496 | 56.75866 |
| 20% | 6353.107 | 4.96533 | 33.76214 | 25.58755 | 7.321344 | 318.0819 | 56.01222 |
| 30% | 6391.733 | 4.96448 | 32.76918 | 23.4578 | 7.035012 | 302.3338 | 56.58238 |
| 40% | 6473.689 | 4.62777 | 32.89412 | 23.7237 | 7.080471 | 307.7371 | 57.10427 |

**Table 4. Imputation effect of DA algorithm under different missing rates.**

| Missing position | Volume | Weather | TR | SR | Shipwreck | PR | Death |
|---|---|---|---|---|---|---|---|
| No missing | 6438.99 | 5.44135 | 37.1087 | 31.0058 | 8.22563 | 353.157 | 56.8793 |
| Pattern 1 | 6248.67 | 5.50659 | 42.1719 | 31.6947 | 8.31577 | 388.333 | 61.6315 |
| Pattern 2 | 6199.59 | 4.97602 | 39.8997 | 32.1348 | 8.87952 | 379.636 | 63.7928 |
| Pattern 3 | 6650.75 | 5.67263 | 41.0402 | 32.8023 | 9.66725 | 356.211 | 73.6988 |
| Pattern 4 | 7157.66 | 5.56097 | 37.2246 | 26.4052 | 7.09627 | 383.821 | 63.2812 |

**Table 5. Imputation effect of EMB algorithm under different missing rates.**

| Missing position | Volume | Weather | TR | SR | Shipwreck | PR | Death |
|---|---|---|---|---|---|---|---|
| No missing | 6438.993 | 5.44135 | 37.10876 | 31.0057 | 8.225633 | 353.1576 | 56.87932 |
| Pattern 1 | 6142.507 | 5.33386 | 37.34791 | 29.3131 | 8.093085 | 352.3846 | 56.98570 |
| Pattern 2 | 6359.274 | 5.38581 | 37.19401 | 31.0881 | 8.126170 | 351.9286 | 56.85189 |
| Pattern 3 | 6293.735 | 5.38469 | 36.32225 | 30.6537 | 8.020468 | 339.7555 | 56.88582 |
| Pattern 4 | 6607.098 | 5.39605 | 36.25956 | 29 | 7.799361 | 339.3866 | 56.69991 |

Volume data, for example, there was more high-value data in missing position pattern 3 and 4, and the standard deviation of the imputation dataset was significantly different from the original. This difference in the effect of imputation at different missing positions was reflected among different data types. Table 5 shows the imputation results of the EMB algorithm. Here, the standard deviation of the dataset imputed by the EMB algorithm was closer to the original dataset at different missing positions. Taking Volume data as an example, even in the two patterns of missing position 3 and 4, the EMB algorithm maintained a good imputation effect. This indicated that the EMB algorithm could better restore the distribution characteristics of data at different missing position, and was less affected by missing positions.

To evaluate the credibility of the dataset interpolated by the EMB algorithm, the probability density curve of the interpolated data and original data, as well as with overimputation

diagnostics were used. Figs 4 and 5 show the relative density curves and overimputation diagnostics of the datasets obtained by different types of data imputation under different missing rates. Due to space limitations, this paper only lists the two variables of TR and Death related to the water safety situation.

The probability density curve in Fig 4 reflects the relationship between the distribution of interpolated mean and original values at different missing rates. The distribution of imputed values under different missing rates was essentially consistent with the distribution of the original values. The main difference was that the ratio of the mode in the distribution of the interpolated mean of the distress number dataset was higher than the original value distribution; that is, the peak value of the original value probability distribution curve was lower. Further, the high value distribution in the original value was relatively average, which caused the posterior distribution generated by the EMB algorithm different from the original distribution in the imputation process, so that the partial imputation value was higher.

However, the proportion of the mode in the mean value distribution of Death was lower than the original value distribution; that is, the peak of the original value probability distribution curve was higher. There were some high peaks in the original values, and such features were not captured by the EMB algorithm, resulting in some extremely high values not being simulated. In general, the distribution of the original value was consistent with the mean value of the imputation, which indicated that the imputation process did not introduce much information or distort the distribution of the original value, but reproduced the original value of the actual distribution. The imputed value was a good substitute for the original value.

Fig 5 shows the overimputation diagnostic results of the distress number dataset at different missing rates. Overimputation means that each original value used in the imputation dataset was set to a missing value one-by-one and multi-interpolated to obtain an imputation mean value and a value confidence interval level of 90%. If the data obtained by the imputation could fully satisfy the random missing mode and normal distribution assumptions, then all the scatter points in Fig 5 will fall on the straight line with a slope of 1 and pass through the origin, but in practice such a situation was often affected by random factors and thus deviates. It is generally believed that as long as the confidence interval level of 90% can cover this aforementioned line, the imputation is considered to be satisfactory. This evaluation method determined whether the missing value imputation could effectively replace the true value. Fig 5 reflects that the confidence intervals of the imputed values at different missing rates mostly cover the theoretical straight line. Only the imputation results of extremely high values were smaller than the original values, which indicated that the EMB algorithm could effectively estimate the data under different missing rates.

Fig 6 shows the results of the overimputation diagnostics for different dataset types at a 10% missing rate. At the 10% missing rate, the 90% confidence level of the overimputation after the imputed dataset essentially covered the straight line with a slope of 1. However, in the three imputed datasets of Volume, PR and Death, there were individual, extremely high values of the imputation results deviating from the original value. Further, the EMB algorithm could better reflect the original distribution of different datasets, but there were still some shortcomings for the simulation of extremely high values. From the overimputation results of dataset on SR, PR and Shipwrecks, the coverage of the straight line with a slope of 1 was very close to 100% for the 90% confidence interval, and the correlation among the three datasets was also reflected in the consistency of the distribution of the overimputation results. Moreover, the EMB algorithm could reflect the correlation between different datasets, and the imputation effect of the dataset without extremum or mutation value was more significant.

The probability density curve in Fig 7 reflects the relationship between the mean distribution of the imputed data and the original data at different missing positions under the 10%

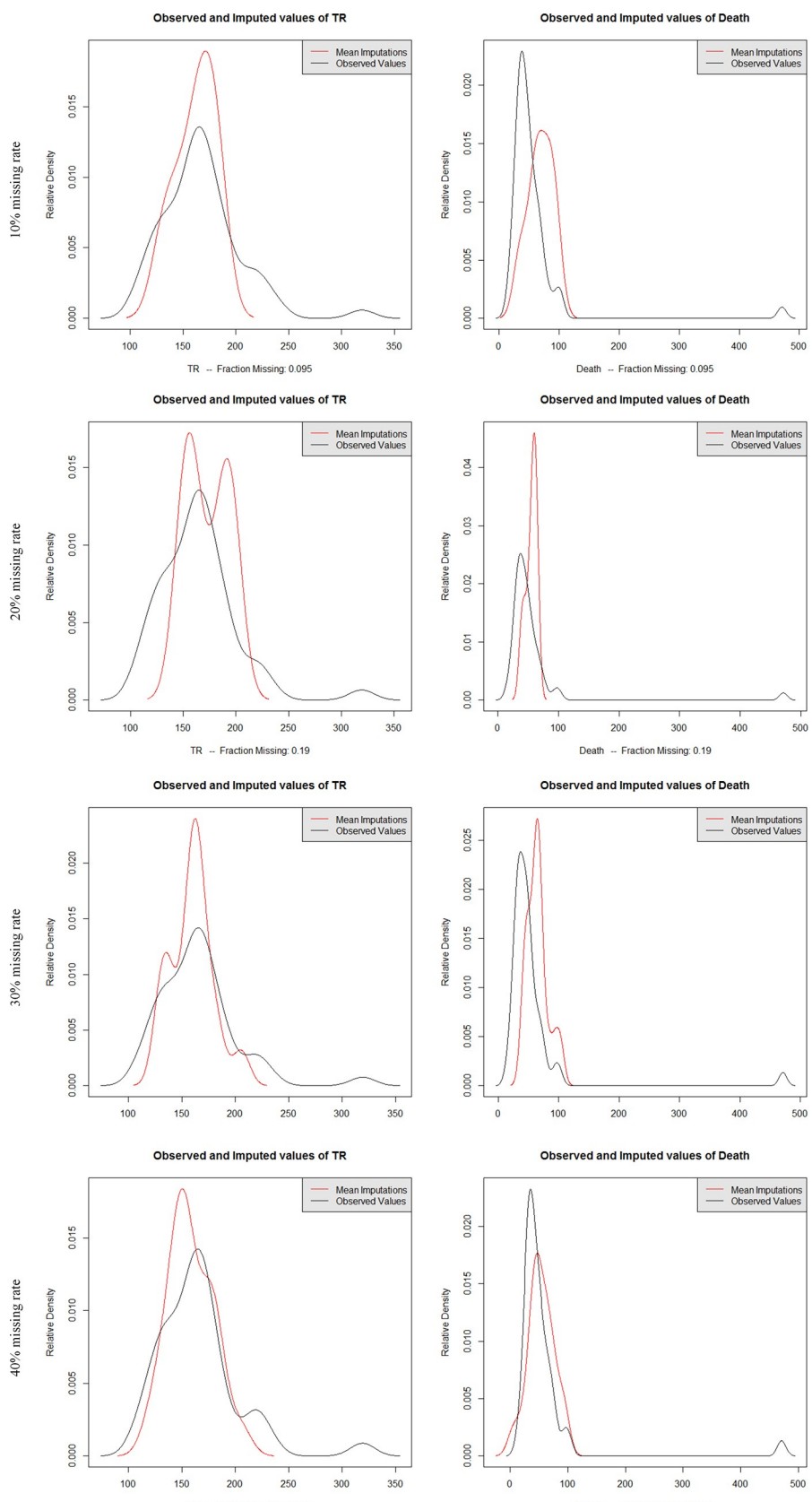

**Fig 4. Probability density distribution of imputed datasets with EMB algorithm under different missing rates.**

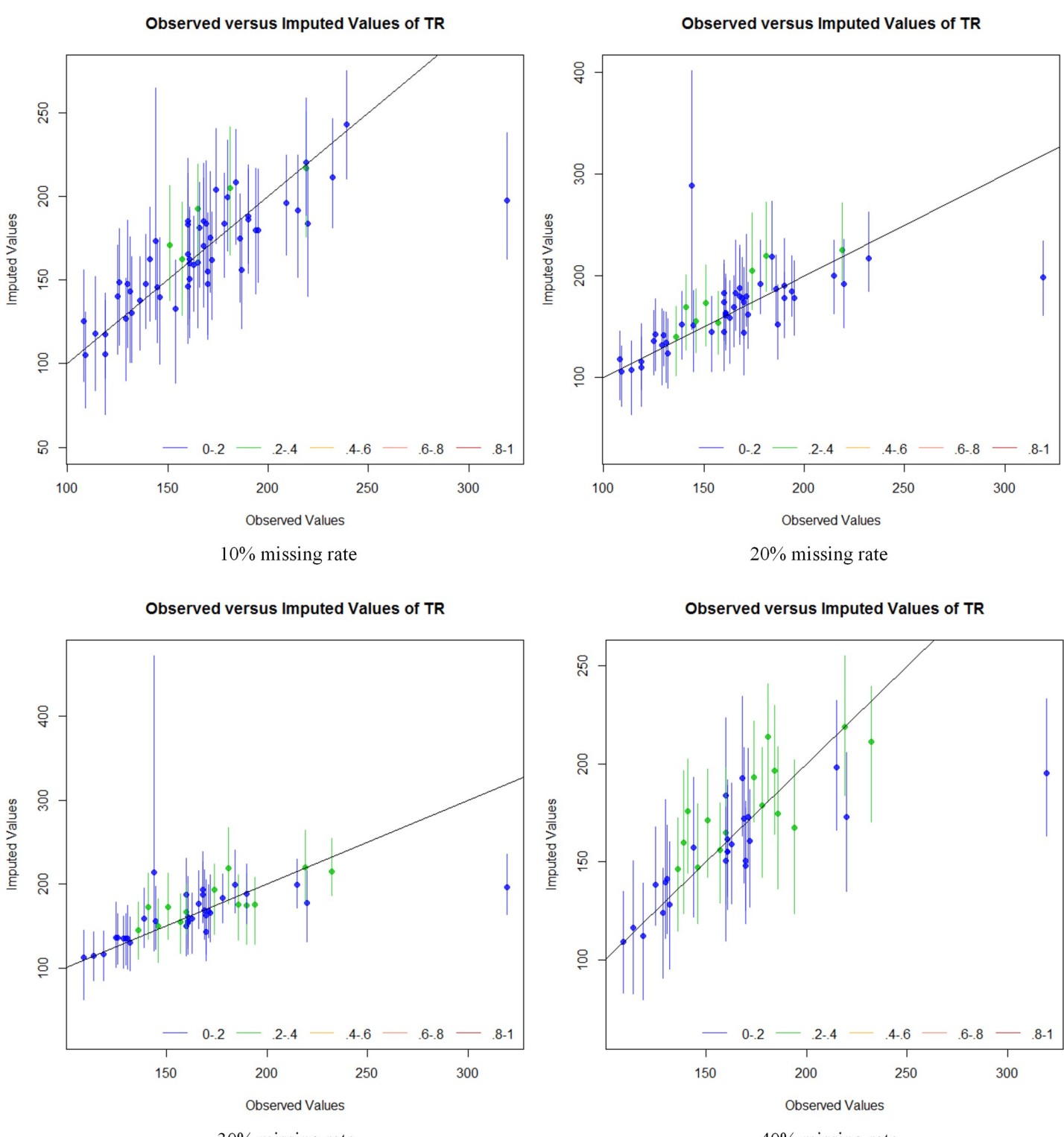

10% missing rate

20% missing rate

30% missing rate

40% missing rate

**Fig 5. Overimputation diagnostics of imputed datasets with EMB algorithm under different missing rates.**

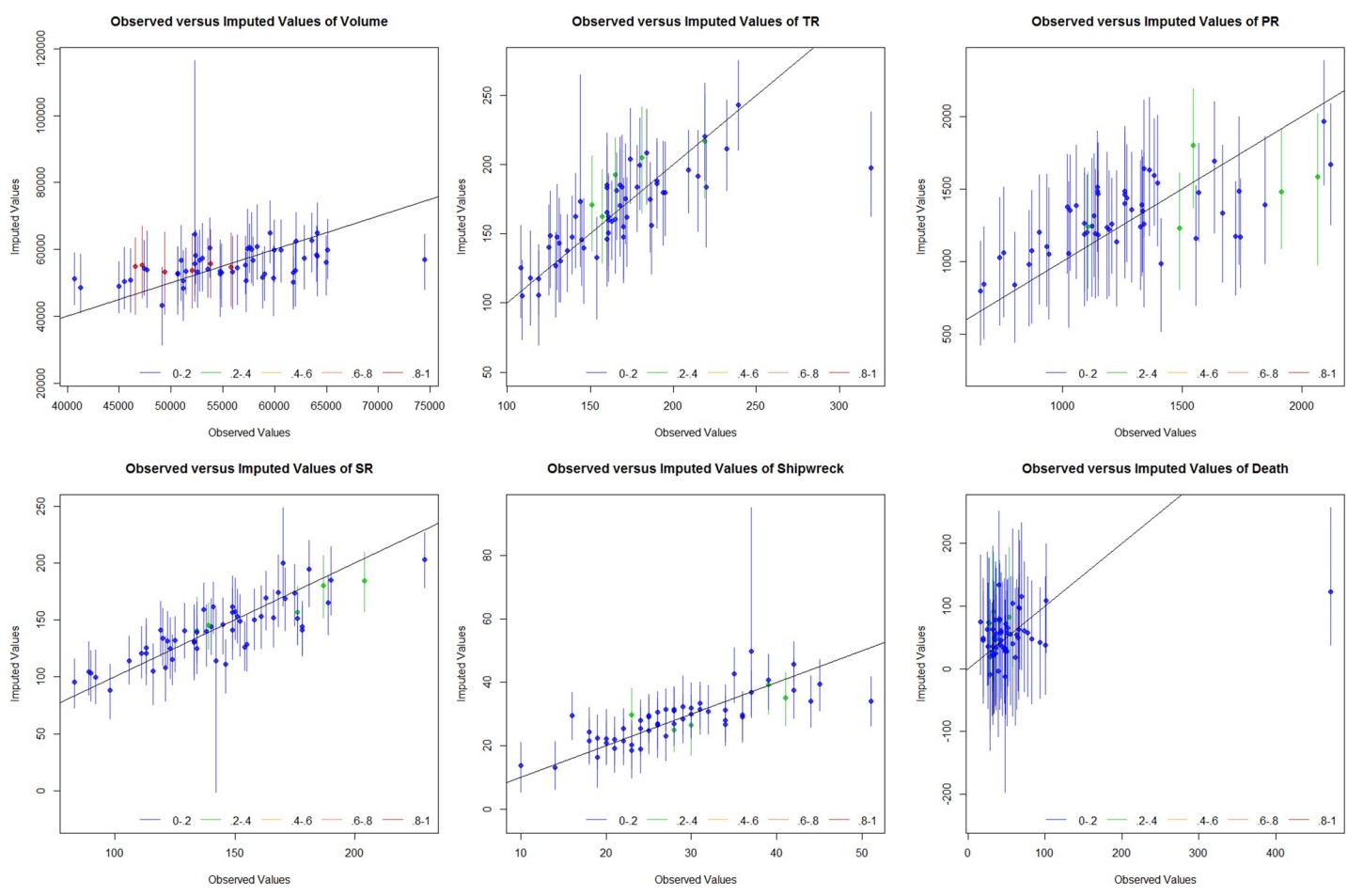

**Fig 6. Overimputation diagnostics of imputed datasets with EMB algorithm under different data types.**

missing rate. Fig 7 shows that the distribution of the dataset obtained by imputation at different missing positions was essentially consistent with the distribution of the original data. For different missing positions, the mode distribution of the missing dataset is different from original dataset, which causes imputed peaks fail to fit better. Due to the various distribution of different data types, the impact of missing positions on imputation results is distinct. In the missing position pattern 1, the missing part of TR and Death contains many high values from across the entire dataset, which leads to a certain deviation of the probability density curve. Especially in the Death dataset, the missing part contains peak data, which causes the peak of the probability density curve to shift backward. It is the same for the distribution of imputed data of Death in missing position pattern 3. In summary, the distribution of the imputed mean values at different positions was consistent with the original value distribution, which showed that the EMB algorithm could better reflect the data distribution characteristics at different positions.

Fig 8 shows the results of the overimputation diagnostic of TR at different missing positions. The confidence interval of the imputed value at different missing positions could essentially cover the theoretical straight line and the imputation result of only one peak data did not reach the original value. Therefore, the EMB imputation method was less affected by the missing positions of different data, and could be effectively applied to the imputation of maritime SAR data. However, EMB imputation effect on data, which is much larger than the mean in

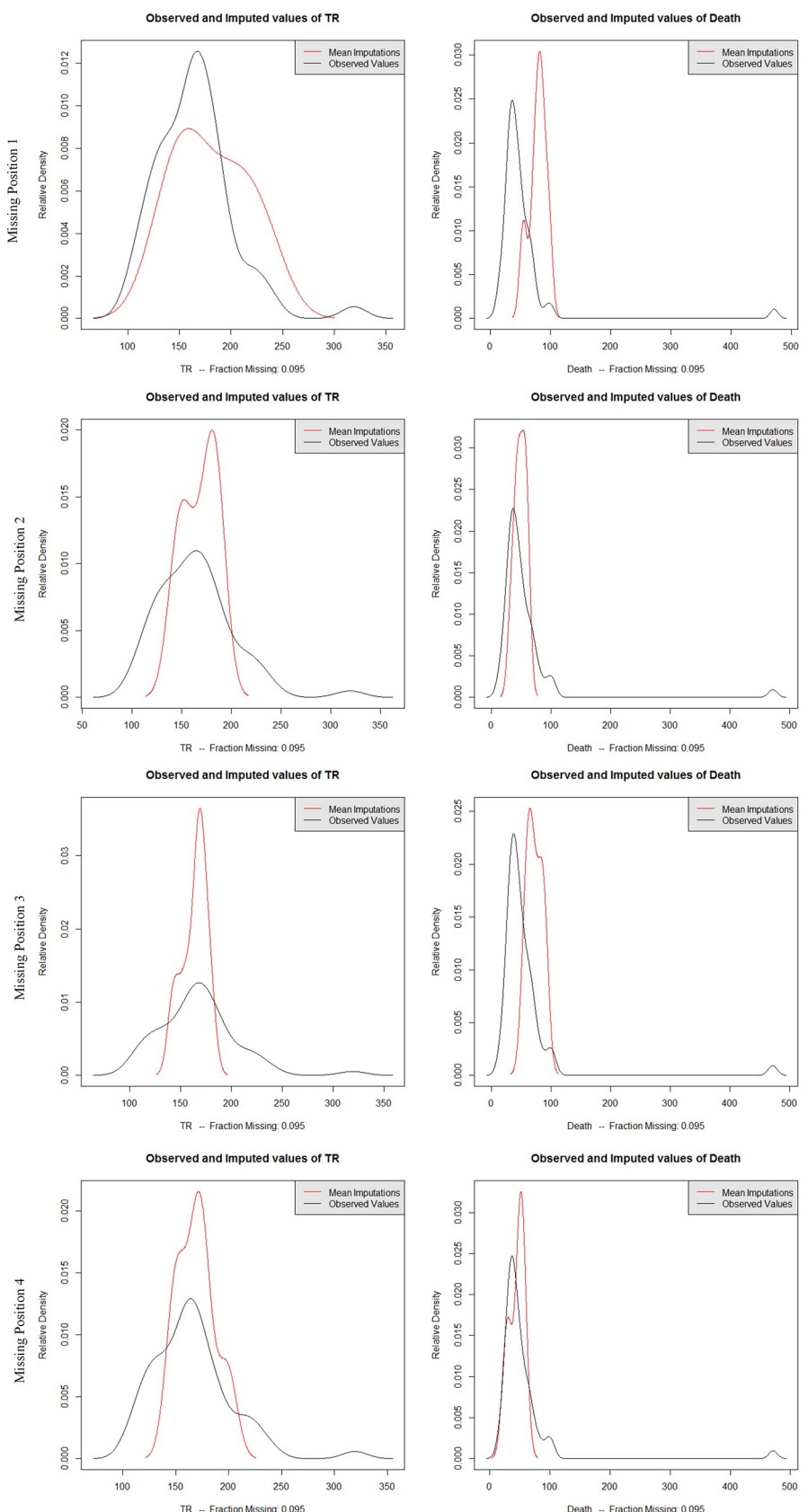

**Fig 7. Probability density distribution of imputed datasets of EMB algorithm under different missing positions.**

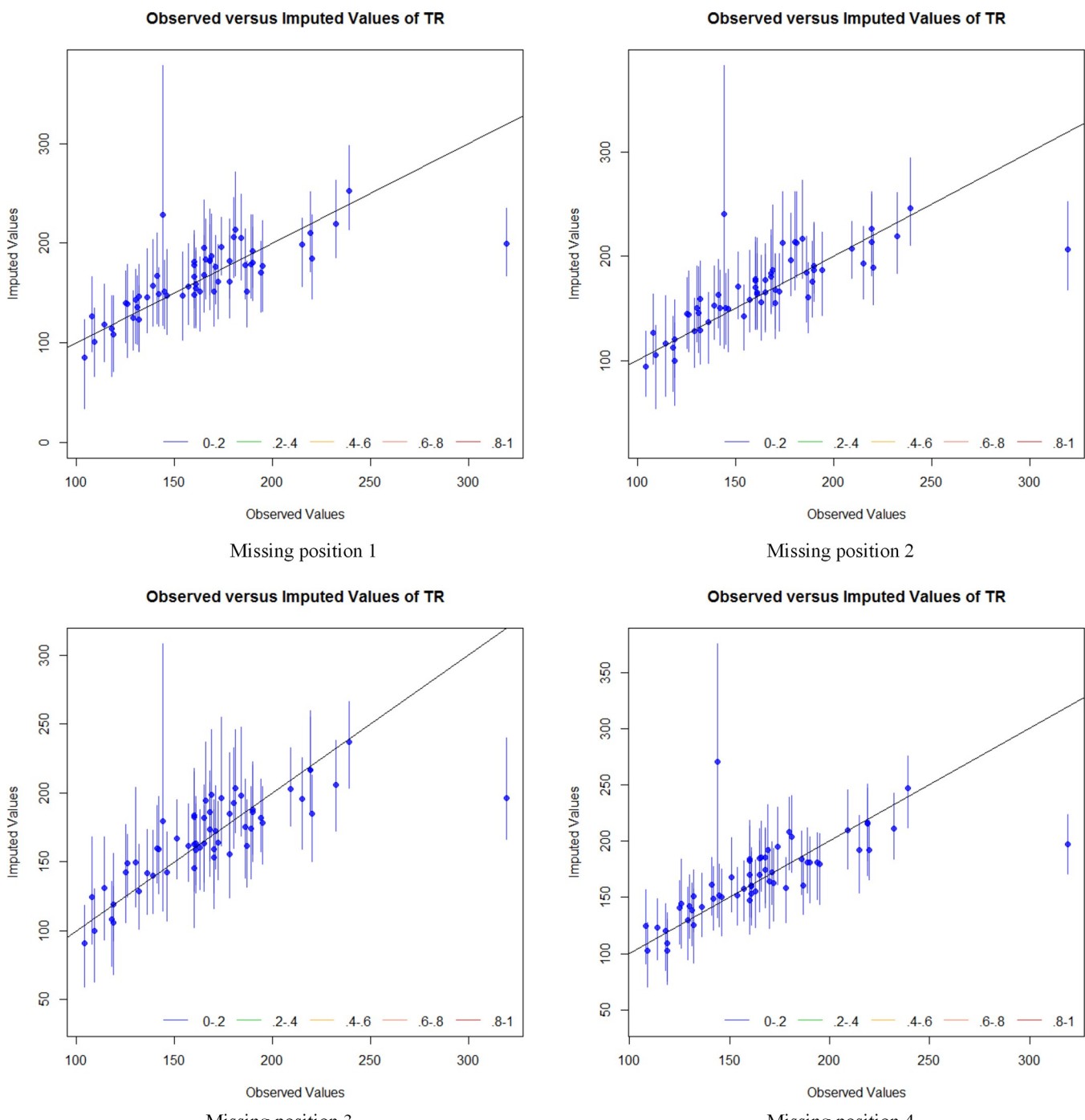

**Fig 8. Imputation results of imputed datasets of EMB algorithm under different missing positions.**

the dataset, is not obvious, and the variability characteristics in the dataset are not restored well. Therefore, before using the EMB algorithm for multiple imputation of the dataset, the variability of the dataset needs to be classified and analysed to better restore the distribution characteristics of the dataset.

## 4 Conclusion

The DA algorithm and EMB algorithm were used to interpolate datasets of different data types at different missing rates and missing positions. Probability density curves and overimputation diagnostics of interpolated datasets revealed that the DA algorithm had a higher computational efficiency and a better imputation effect on the dataset with a lower missing rate. Compared with the DA algorithm, the interpolated data obtained by the EMB algorithm could substantially restore the actual distribution of the dataset. The EMB algorithm was also less affected by different data missing positions. In addition, the dataset obtained by imputation with a high data missing rate was still highly credible, and EMB algorithm could reflect the correlation among different datasets. The imputation effect was more significant for the dataset without extremum or a mutation value. The EMB algorithm has better application and promotion significance.

However, the imputation effect of the EMB algorithm on the extreme data was poor, and the variability characteristics of the dataset could not be restored well. Further data analysis based on the characteristics of extreme or mutated data is necessary to improve the data imputation effect, provide data support for maritime SAR analysis, prevent accidents and disasters effectively, and improve the safety level continuously in waterborne transportation.

## Author Contributions

**Conceptualization:** Fengyun Chen, Liansheng Xu.

**Data curation:** Guobo Wang.

**Formal analysis:** Minglu Ma.

**Investigation:** Minglu Ma.

**Project administration:** Lili Jiang.

**Resources:** Liansheng Xu.

**Software:** Lili Jiang.

**Supervision:** Liansheng Xu.

**Validation:** Fengyun Chen.

**Writing – original draft:** Guobo Wang.

**Writing – review & editing:** Guobo Wang.

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
