## [Decision Letter · Decision Letter 0]

19 Nov 2020

PONE-D-20-14450

Multiple imputation of maritime search and rescue data at multiple missing patterns

PLOS ONE

Dear Dr. JIANG,

Thank you for submitting your manuscript to PLOS ONE. After careful consideration, we feel that it has merit but does not fully meet PLOS ONE’s publication criteria as it currently stands. Therefore, we invite you to submit a revised version of the manuscript that addresses the points raised during the review process.

We look forward to receiving your revised manuscript.

Kind regards,

Lubos Buzna, Ph.D

Academic Editor

PLOS ONE

Journal Requirements:

2.We suggest you thoroughly copyedit your manuscript for language usage, spelling, and grammar. If you do not know anyone who can help you do this, you may wish to consider employing a professional scientific editing service.  

Additional Editor Comments (if provided):

Two of our reviewers provided reports on your paper raising some concerns about its publication. Please consider revision of the paper, while considering their comments. If you decide to resubmit the paper, please prepare a detailed description of amendments and reply to comments.

Reviewers' comments:

Reviewer's Responses to Questions

**Comments to the Author**

1. Is the manuscript technically sound, and do the data support the conclusions?

Reviewer #1: Partly

Reviewer #2: Partly

2. Has the statistical analysis been performed appropriately and rigorously? 

Reviewer #1: No

Reviewer #2: Yes

3. Have the authors made all data underlying the findings in their manuscript fully available?

Reviewer #1: Yes

Reviewer #2: Yes

4. Is the manuscript presented in an intelligible fashion and written in standard English?

Reviewer #1: Yes

Reviewer #2: No

5. Review Comments to the Author

Reviewer #1: This paper uses multiple imputation methods to deal with the missing data of marine search and rescue, and verifies the effect of multiple imputation in different missing patterns, but the article lacks novelty and general significance.

1.The authors do not reference original sources, such as seminal work by Rubin done in 1970's on multiple imputation.

2.This article compares the effect of DA algorithm and EMB algorithm on the processing of missing data, but it does not highlight the particularity of maritime search and rescue data.

3.Please clarify your significant contributions besides utilizing existing algorithms for handling missing data.

4.It has to be described if the algorithm used in this article is better than data imputation based on professional experience in the field of maritime search and rescue.

5.The article only imputation the missing data within five years. Will it affect the results in the case of a long time series?

Here are some clerical mistakes:

1.Delete “based on” on line 45.

2.Irregular chart format in table 5.

3.The figures are not clear.

Reviewer #2: Well known multiple imputation method and algorithms are properly applied. Data and results clearly presented in figures.

1.In case of some variables the general sense of applying this method for SAR purposes rises questions and reservations:

1) “Death” (“Death is the number of missing or dead persons”).

What is a purpose of assessing (or receiving) number of “death” (in previous years) by statistical methods?

Who (or what kind of institution) is interested in receiving such data?

2) “Weather” (“Weather is the number of bad weather”)

To be defined what is “bad weather” – e.g. wind force higher than 8° B, 10°B, sea state, tropical storms, poor visibility?

Why do the Authors try to input missing data by statistical methods instead of checking real meteorological surface analysis existing in the shipping world and other meteorological data bases?

3) “Volume” (“Volume is the volume of waterborne transport”)

Figure 6 . Observed versus imputed Value of Volume -

What are the units? ( e.g. 75000 - individual ships observed/ analyzed?)

4) “TR” (“TR is the times of distress”)

Units again.

E.g. What is TR 200? (Figure 6)

2.Other remarks:

1) Table 1. “Safety status” to be used rather than “Security status”, as the word “Security” is used in shipping for piracy and maritime terrorism issues.

2) Figure 4 - axes x - missing labels at 30% missing rate.

3) Generally clear and understandable language, but some sentences require language corrections (including professional maritime vocabulary) as they are difficult to understand (e.g. In 2018, China had a total of 176 water traffic accidents with general and higher levels associated with transportation ships, causing 237 people missing, 83 shipwrecks, and a direct economic loss of 290 million yuan [4-5].)

6. PLOS authors have the option to publish the peer review history of their article (what does this mean?). If published, this will include your full peer review and any attached files.

Reviewer #1: No

Reviewer #2: No

---

## [Author Response · Author response to Decision Letter 0]

3 Jan 2021

Response to Reviewer #1: Thank you for your valuable comments. The amendments to your comments are as followings:

1. The authors do not reference original sources, such as seminal work by Rubin done in 1970's on multiple imputation.

I have read many papers on multiple imputation by Rubin, such as Inference and missing data published in 1976, and Statistical analysis with missing data published in 2002, which created a precedent for data imputation. Actually, my paper is mainly about the latest applications of multiple imputation, and citation of original sources would also be helpful. Rubin's Inference and missing data published in 1960 has already been cited in the paper.

2. This article compares the effect of DA algorithm and EMB algorithm on the processing of missing data, but it does not highlight the particularity of maritime search and rescue data.

This paper has already explained the composition of search and rescue data when introducing the data source, and specifically analyzed the missing features of search and rescue data when constructing the missing model, and these are the main conditions required for data imputation.

3. Please clarify your significant contributions besides utilizing existing algorithms for handling missing data.

The main contribution of this paper is to analyze the characteristics of search and rescue data, and adopt a relatively reliable method to impute the data based on the comparison of interpolation methods, which better solves the problem of data missing in search and rescue simulation. Besides that, evaluation methods such as probability density curves and over-imputation diagnostics are used to evaluate the effect of data interpolation more accurately, which evaluate the effect of interpolation depending on the data distribution, and avoid erroneous evaluation of the effect of interpolation by averaging methods such as interpolation average. This article provides new ideas for method selection and effect evaluation of data interpolation.

4. It has to be described if the algorithm used in this article is better than data imputation based on professional experience in the field of maritime search and rescue.

Due to the difference among areas, water environments, personnel qualities and other factors, there is great uncertainty in the number of ships in distress. Consequently, it seems that the method of data interpolation using professional experience was not applied at present. This also explains the rare usage of data interpolation in the field of maritime search and rescue.

5. The article only imputation the missing data within five years. Will it affect the results in the case of a long time series?

Data volume is sufficient to meet the demands of multiple imputation sample construction. It will not affect the results in the case of a long time series.

Here are some clerical mistakes:

1. Delete “based on” on line 45.

"Based on" in line 45 has been deleted.

2. Irregular chart format in table 5.

The format in Table 5 has been adjusted.

3. The figures are not clear.

The picture meets the resolution requirement, and it is clear, even after a click to enlarge. If it still cannot be displayed clearly, I will upload the picture again.

Response to Reviewer #2: Thank you for your constructive amendments. The amendments to your comments are as follows:

1. In case of some variables the general sense of applying this method for SAR purposes rises questions and reservations:

1) “Death” (“Death is the number of missing or dead persons”).

What is a purpose of assessing (or receiving) number of “death” (in previous years) by statistical methods? 

The main purpose of counting the death toll is to quantify the risk of accidents, and it also reflects the difficulty of search and rescue and the level of water traffic safety.

Who (or what kind of institution) is interested in receiving such data?

Government departments are more interested in these data, especially maritime management departments. These data help them understand the current difficulty of water search and rescue and whether they can complete the search and rescue mission. 

2) “Weather” (“Weather is the number of bad weather”)

To be defined what is “bad weather” – e.g. wind force higher than 8° B, 10°B, sea state, tropical storms, poor visibility?

The number of bad weather in this article specifically refers to the number of days affected by typhoons, because typhoon has the greatest impact on the safety of ships at sea. Futherer, typhoon is also a comprehensive variable, containing the factors of rain, visibility, wind, etc.

Why do the Authors try to input missing data by statistical methods instead of checking real meteorological surface analysis existing in the shipping world and other meteorological data bases?

The data used to analyze the missing pattern in this article is the actual missing data set, and the subsequent data set used for the missing analysis is the complete data to analyze the accuracy of the imputation result. Therefore, the subsequent missing analysis of meteorological data is to evaluate the effect of imputation, and will not replace the existing data with imputed data.

3) “Volume” (“Volume is the volume of waterborne transport”)

Figure 6 . Observed versus imputed Value of Volume -

What are the units? ( e.g. 75000 - individual ships observed/ analyzed?)

The abscissa is the actual waterway freight volume of the month, its unit is Ton. The ordinate is the interpolation data of the observed value, its unit is also Ton.

4) “TR” (“TR is the times of distress”)

Units again. 

E.g. What is TR 200? (Figure 6)

The abscissa is the actual times of distress in month, its unit is Time. The ordinate is the interpolation data of the observed value, its unit is also Time. TR 200 means the distress of this month is 200 times.

Other remarks:

1) Table 1. “Safety status” to be used rather than “Security status”, as the word “Security” is used in shipping for piracy and maritime terrorism issues.

Agree with the amendment and change "Security status" to "Safety status".

2) Figure 4 - axes x - missing labels at 30% missing rate.

Agree with the amendment and add the labels at 30% missing rate.

3) Generally clear and understandable language, but some sentences require language corrections (including professional maritime vocabulary) as they are difficult to understand (e.g. In 2018, China had a total of 176 water traffic accidents with general and higher levels associated with transportation ships, causing 237 people missing, 83 shipwrecks, and a direct economic loss of 290 million yuan [4-5].)

Agree with the amendment and modify the sentence in the paper.

---

## [Decision Letter · Decision Letter 1]

11 May 2021

Multiple imputation of maritime search and rescue data at multiple missing patterns

PONE-D-20-14450R1

Dear Dr. JIANG

We’re pleased to inform you that your manuscript has been judged scientifically suitable for publication and will be formally accepted for publication once it meets all outstanding technical requirements.

Kind regards,

Lubos Buzna, Ph.D

Academic Editor

PLOS ONE

Additional Editor Comments (optional):

Both reviewers recognized that the paper has been significantly improved and do not have any additional significant comments. I suggest authors to check remaining minor comments of Reviewer 1, in particular those suggesting some additional explanations.

Reviewers' comments:

Reviewer's Responses to Questions

**Comments to the Author**

1. If the authors have adequately addressed your comments raised in a previous round of review and you feel that this manuscript is now acceptable for publication, you may indicate that here to bypass the “Comments to the Author” section, enter your conflict of interest statement in the “Confidential to Editor” section, and submit your "Accept" recommendation.

Reviewer #1: All comments have been addressed

Reviewer #2: All comments have been addressed

2. Is the manuscript technically sound, and do the data support the conclusions?

Reviewer #1: Yes

Reviewer #2: Yes

3. Has the statistical analysis been performed appropriately and rigorously? 

Reviewer #1: Yes

Reviewer #2: Yes

4. Have the authors made all data underlying the findings in their manuscript fully available?

Reviewer #1: Yes

Reviewer #2: Yes

5. Is the manuscript presented in an intelligible fashion and written in standard English?

Reviewer #1: Yes

Reviewer #2: Yes

6. Review Comments to the Author

Reviewer #1: 1. The label of the article formula should be aligned as much as possible

2. The calculation process of the EMB algorithm in Figure 1 needs to be explained in more detail, including the meaning of different colors and different graphic annotations in the figure, etc.

3. The units of the parameter variables in Table 2 to Table 5 need to be explained

4. In the article, “Weather” represents the number of bad weather, and the meaning or indicators of bad weather need to be explained

Reviewer #2: The article presents very valuable research results. All reservations were clearly explained. All questions were answered to my satisfaction. Small were corrections done as indicated.

7. PLOS authors have the option to publish the peer review history of their article (what does this mean?). If published, this will include your full peer review and any attached files.

Reviewer #1: No

Reviewer #2: **Yes: **Miroslaw Wielgosz

---

## [Editor Report · Acceptance letter]

11 Jun 2021

PONE-D-20-14450R1 

Multiple imputation of maritime search and rescue data at multiple missing patterns 

Dear Dr. Jiang:

I'm pleased to inform you that your manuscript has been deemed suitable for publication in PLOS ONE. Congratulations! Your manuscript is now with our production department. 

Kind regards, 

on behalf of

Prof. Lubos Buzna 

Academic Editor

PLOS ONE